# Structure and Dynamics of Spider Silk Studied with Solid-State Nuclear Magnetic Resonance and Molecular Dynamics Simulation

**DOI:** 10.3390/molecules25112634

**Published:** 2020-06-05

**Authors:** Tetsuo Asakura

**Affiliations:** Department of Biotechnology, Tokyo University of Agriculture and Technology, 2-24-16 Nakacho, Koganei, Tokyo 184-8588, Japan; asakura@cc.tuat.ac.jp; Tel.: +81-42-383-7733

**Keywords:** solid-state NMR, molecular dynamic simulation, spider dragline silk, structure, dynamics

## Abstract

This review will introduce very recent studies using solid-state nuclear magnetic resonance (NMR) and molecular dynamics (MD) simulation on the structure and dynamics of spider dragline silks conducted by the author’s research group. Spider dragline silks possess extraordinary mechanical properties by combining high tensile strength with outstanding elongation before breaking, and therefore continue to attract attention of researchers in biology, biochemistry, biophysics, analytical chemistry, polymer technology, textile technology, and tissue engineering. However, the inherently non-crystalline structure means that X-ray diffraction and electron diffraction methods provide only limited information because it is difficult to study the molecular structure of the amorphous region. The most detailed picture of the structure and dynamics of the silks in the solid state experimentally have come from solid-state NMR measurements coupled with stable isotope labeling of the silks and the related silk peptides. In addition, combination of solid-state NMR and MD simulation was very powerful analytical tools to understand the local conformation and dynamics of the spider dragline silk in atomic resolution. In this review, the author will emphasize how solid-state NMR and MD simulation have contributed to a better understanding of the structure and dynamics in the spider dragline silks.

## 1. Introduction

There are a variety of silkworms and spiders, each producing silk with unique primary and higher order structures [1,2]. Over nearly 400 million years, orb-web weaving spiders have evolved the ability to produce up to six different silks and one glue-like substance that possesses amazing, yet unique, mechanical properties as summarized in Figure 1 [2,3,4,5]. Among the spider silks, spider dragline silks have very excellent mechanical properties such as the combination of high strength and high toughness [1,2,3,4,5,6,7]. Another unique characteristic of the spider silks is supercontraction, which occurs in the process of hydration. Interaction with water causes spider dragline silk fiber to contract up to 50% in length and swell in diameter [8,9,10]. In addition, recent developments in biotechnology have opened the possibility to reproduce the recombinant spider silk proteins [11]. These appealing backgrounds have attracted researchers in diverse fields, such as biology, biochemistry, biophysics, analytical chemistry, polymer technology, textile technology, and biomaterials. The unique properties of spider silks are essentially originated from the structure and dynamics, and therefore, the atomic resolution information on the structure gives us the answer to why the silk has such unique and excellent properties [1,7,12].

A variety of techniques such as X-ray diffraction, Fourier transform infrared/Raman spectroscopies, and transmission electron microscopy have been applied to clarify the structure of the spider dragline silk [13,14,15,16,17,18,19,20,21,22,23,24,25]. By using these techniques, the structures have been studied from secondary structures to molecular arrangement to hierarchical structure. For example, the X-ray diffraction technique has been used extensively to clarify the structure of silk at the atomic/molecular level, including the size and orientation of nano-crystallites, and also molecular chain arrangement. However, because of the semi-crystalline nature of silk, it is quite difficult to study the atomic/molecular structure of the amorphous region in the silk by X-ray diffraction. To replace or complement this technique, NMR spectroscopy has been applied recently and has been demonstrated as a very effective method to clarify the structure and dynamics of the dragline spider silk [26,27,28,29,30,31,32,33,34,35,36,37,38,39,40,41,42,43,44,45,46,47,48,49,50,51,52,53,54,55,56,57,58,59,60,61,62,63,64,65]. The solid-state NMR studies used a combination of approaches [66]. Spiders can be fed on an isotopically labeled diet, allowing isotopic labeling of silk within the silk gland: either uniformly or by feeding amino acids labeled at individual positions [27,28,29,30,31,32,34,35,36,37,40,42,43,44,45,47,48,49,50,53,54,55,56,59,61]. In addition, synthetic peptides with isotopically labelings in specific positions have proved very useful model systems permitting site-specific structural information [33,38,39,41,46,51,52,60,62,63,64,65]. The conformation-dependent ^13^C chemical shift in the ^13^C cross polarization and magic angle spinning (CP/MAS) NMR spectra of silks and the model peptides coupled with selective ^13^C labeling can be used effectively for determination of local structure at secondary structure level in an amino acid-specific manner [66,67,68,69,70,71,72]. In particular, the combination of ^13^C labeling of specific sites in amino acids, and ^13^C Cα and Cβ chemical shift contour maps of these sites is very useful for the structural study. Moreover, many kinds of advanced solid-state NMR techniques have been applied to obtain local structure and dynamics of spider silk at atomic resolution as will be shown below [27,28,29,30,31,32,33,34,35,36,37,38,39,40,41,42,43,44,45,46,47,48,49,50,51,52,53,54,55,56,61].

On the other hand, molecular dynamics (MD) simulation is very effective to understand the local conformation and dynamics of *Nephila clavipes* (*N. clavipes*) dragline silks theoretically and to verify the experimental results [56,58,62,63,65]. MD simulation can play a critical role in connecting experimental restraints with potentially plausible molecular structures. Therefore, it has been also used to understand the structure and structural changes of silk fibroins including silk fiber formation mechanism and the chain dynamics [73,74,75,76,77,78,79,80,81,82,83,84,85]. Especially, combination of solid-state NMR and MD simulations was very strong analytical tools to understand the local conformation and dynamics of the spider dragline silk in atomic resolution [56,58,62,63,65].

We previously reviewed about NMR studies of silk several times [86,87,88,89,90,91] and most recently, an excellent review about NMR characterization of silk including spider silk was published by Guo and Yarger [92]. Therefore, the author hopes to avoid duplication of contents here.

In this review, recent studies about the structure and dynamics of the dragline silks of two kinds of spiders, *N. clavipes* and *Nephila clavata* (*N. clavata*) in the dry and hydrated states using solid-state NMR and MD simulation were introduced because the dragline silk of the golden-orb weaver, *N. clavipes*, has become the benchmark and *N. clavata* can be easily obtained in Japan.

## 2. Major Ampullate Silk Fiber of *N. clavipes* and Their ^13^C Solid-State NMR Spectra

Spider dragline silks are synthesized in ampullate glands located in the abdomen and spun through a series of orifices (spinnerets) [2,3,4,5]. The silk fiber originating in the major ampullate gland has been the most studied. The major ampullate silk consists of two structural proteins, designated as spidron 1 (MaSp1) and spidron 2 (MaSp2) [93]. The dominant MaSp1 protein can be described as a block copolymer consisting of polyalanine (poly-A) domain as the crystalline region and Gly-rich regions as the less structurally ordered non-crystalline region [1,2,3,4,5,6,7]. The former region has been associated with high fiber strength, and the latter is a source of the high elasticity observed for the spider silk fiber. They assemble into fibers with the excellent mechanical properties.

The dragline silk of *N. clavipes* has become the benchmark and therefore, let us start by introducing NMR studies of this spider dragline silk. The ^13^C CP/MAS NMR spectrum of the major ampullate silk fibers of *N. clavipes* reported by Simmons et al. [26] is shown in Figure 2 together with the assignment.

The integrated areas of peaks from Gly, Ala, Glu, and Tyr were 42%, 46%, 9%, and 4%, compared to values of 43%, 30%, 7%, and 4% obtained by amino acid analysis, which means that both the crystalline and amorphous regions of silk are detected by ^13^C CP/MAS NMR. The Ala Cα and Ala Cβ chemical shifts were 49.0 and 20.6 ppm, respectively, indicating that the spider silk fiber took antiparallel β (AP-β) sheet structure judging from the ^13^C conformation-dependent chemical shift data [67,68,69,70,71,72]. On the other hand, the structural analysis of the Gly-rich region is difficult because the region is heterogeneous and amorphous. Therefore, the combination of solid-state NMR and MD simulation together with use of appropriate stable-isotope labeled model peptides of the typical sequences in the Gly-rich region is very effective to clarify the structure and dynamics.

## 3. Structure of the Gly-Rich Region of *N. clavipes* Dragline Silk

The typical amino acid sequence including poly-A and Gly-rich regions of *N. clavipes* dragline silk reported by Xu and Lewis [4] is partially shown in Figure 3.

It is difficult to analyze the structure of the Gly-rich region directly from simple ^13^C CP/MAS NMR spectrum of the dragline silk as shown in Figure 2 because of the heterogeneity of the primary structure. Instead of this, the use of synthetic peptides with isotopically labelings in specific positions provides a sequential model of the Gly-rich region permitting site-specific structural information. Therefore, the selectively stable-isotope labeled 47-mer peptides flanked by (Ala)_6_ at both ends were synthesized as summarized in Table 1 [46,60,62]. The sequence was selected in Figure 3 as underlined sequence, a and poly-Glu blocks were attached at both N- and C-termini to make them water-soluble [94]. In order to determine the local conformation of Gly-rich region, rotational echo double resonance (REDOR) NMR [29,46,95,96] was used to determine the inter-nuclear distance between the ^15^N and ^13^C labeled sites as summarized in Table 1 (I).

The inter-nuclear distances between ^13^C(i) and ^15^N(i+2) nuclei in the Gly(i)-X-Gly(i+2) motifs were also calculated using the torsion angles of the X amino acid (X = Q, Y, L, and R) taken from the Protein Data Bank (PDB) database [46] and plotted together with the corresponding distances determined by the REDOR experiment in Figure 4 (red circles). The corresponding distances of typical conformations are also shown in Figure 4 (gray hollow circles), i.e., AP-β sheet; 4.6 Å, α-helix and type II β-turn; 3.2 Å and 3_1_-helix; 4.2 Å [46]. The REDOR-determined distances did not match any distances of typical conformations. Therefore, it is concluded that the Gly-rich region took random coil and no well-defined conformation.

However, the presence of 3_1_-helix (Φ and Ψ = −90°, 150°) in the Gly-rich region has been frequently proposed in previous papers [28,32,42,43]. Therefore, it is important to examine the possibility about the presence of 3_1_-helix in detail using these 47-mer peptides.

The ^13^C chemical shift of Ala Cβ peak, 17.4 ppm is a measure of the presence of 3_1_-helix as reported previously [33] because the value, 17.4 ppm is the ^13^C chemical shift of the Ala Cβ peak of poly(Ala-Gly-Gly) with the 3_1_-helix form confirmed by X-ray diffraction analysis [97]. Here it is necessary to be careful about the chemical shift reference when comparing the ^13^C chemical shift-dependent conformations of silks. We used tetramethylsilane, TMS, as the external chemical shift reference in the solution and solid-state NMR of silks [66,70,71]. Three kinds of the 47-mer peptides introduced three successive ^13^C-labeled amino acid residues, [2-^13^C]Gly, [3-^13^C]Ala, and [1-^13^C]Gly at different positions were synthesized as listed in Table 1(II) [62]. The ^13^C CP/MAS NMR spectra of three Ala Cβ peaks of the Ala^13^, Ala^26^, and Ala^36^ residues in the [2-^13^C]Gly[3-^13^C]Ala[1-^13^C]Gly motifs at the different three positions are shown in Figure 5. Judging from the ^13^C chemical shifts of Ala Cα, Cβ, and C = O peaks of (Ala)_6_ residues at the both ends, these residues took on the AP-β sheet structure. It was clear that the Ala^13^, Ala^26^, and Ala^36^ residues in the Gly-Ala-Gly motifs in the Gly-rich region did not adopt the 3_1_-helix because the Cβ chemical shifts of these three Ala residues were not 17.4 ppm, which shows a typical 3_1_-helix. In order to determine the local conformations of the individual Ala and Gly residues in detail, the fractions of individual local conformations were determined from the ^13^C labeled Gly and Ala residues of these three kinds of peptides by the peak deconvolution analysis. In addition, those of other Gly residues were determined for six kinds of the peptides used for REDOR experiments (Table 1 (I)) and two additional 47-mer peptides with [1-^13^C]Gly^12^or [1-^13^C]Gly^35^ residues [62].

The fractions of several conformations were plotted against the position of the ^13^C-labeled nuclei in a 47-mer peptide chain in Figure 6a. Due to the presence of poly-A sequences with AP-β structure at both N- and C-termini, the residues located close to the poly-A sequences tended to adopt the AP-β structure. Actually, the fractions of AP-β structure rapidly decreased toward the center from the edge, in contrast to those of random coil. The fraction of β-turn structure tended to decrease after increasing once towards the center from the edge although the change was relatively small. In order to examine the change in the fractions theoretically, MD simulations were performed as shown in Figure 6b [62]. Thus, the observed results were successfully rationalized through the MD simulation.

## 4. Packing Structure of the Polyalanine Region of *N. clavipes* Dragline Silk

It is important to clarify the packing structure of poly-A region with AP-β sheet structure to clarify the origin of higher fiber strength in the atomic level, but there have been no detailed reports about the packing structure so far. Therefore, the packing structure of poly-A region was started to analyze for a series of alanine oligopeptides, (Ala)_n_ (n = 4–8 and 12) with AP-β sheet structures using both solid-state NMR spectroscopy and X-ray crystallography as shown in Figure 7 [52]. Here (Ala)_12_ is the typical number of the Ala residue used for the model of poly-A region in *Samia cynthia ricini* (*S. c. ricini*) silk fibroin whose primary structure is similar to that of the spider dragline silk [88]. The ^13^C chemical shifts and line shapes of the Ala Cβ carbons were used together with X-ray diffraction patterns for the structural analyses. The atomic co-ordinates of (Ala)_4_ were determined using a single crystal X-ray diffraction analysis of the single crystal (Ala)_4_ sample [52]. It is noted that both the line shapes of the Ala Cβ carbons and X-ray diffraction powder patterns of (Ala)_n_ (*n* = 7, 8, and 12) were markedly different from those for short alanine oligopeptides, (Ala)_n_ (*n* = 4–6). Namely, the chemical shifts of the central peaks with highest intensity were 20.4 ppm and the same among the corresponding chemical shifts of the Ala Cβ carbons of (Ala)_n_ (*n* = 4–6). On the other hand, the Ala Cβ peaks had two peaks at 22.7 ppm and 19.6 ppm together with a broad peak at 21 ppm.

In the X-ray diffraction spectra, the second peak 2 shifted from 2θ = 19.2° to 2θ = 20.4° remarkably [52]. These data indicate that the packing arrangement of alanine oligopeptides, (Ala)_n_ with the AP-β sheet structure changed from rectangular (*n* = 4–6) to staggered (*n* = 7, 8, and 12) arrangement.

Later, the transition from the rectangular to the staggered arrangement in (Ala)_6_ was observed from the change in the Ala Cβ peak through heat treatment at 200 °C for 4 h and therefore the rectangular structure was stable for (Ala)_6_ and shorter poly-A chains when the stabilization of the intermolecular hydrogen bonding with bound water molecules occurred at the end groups of the terminal residues [58]. However, poly-A regions existed between both ends of Gly-rich sequences in spider dragline silks and therefore, the packing states of poly-A regions should be examined by taking into account these sequences again.

In Figure 3, the underlined sequence (b) was selected and a series of the sequential peptides where the poly-A regions exist between both ends, i.e., (E)_4_GGLGGQGAG(A)_n_GGAGQGGYGG(E)_4_ (*n* = 3–8) were studied for analyzing the packing structure of the poly-A region. The Gr(A)_n_Gr* (*n* = 3–8) samples were soluble in water because of the presence of two (Glu)_4_ blocks at both termini. Here, Gr = (E)_4_GGLGGQGAG and Gr* = GGAGQGGYGG(E)_4_. After lowering the pH to 2, the Gr(A)_n_Gr* (*n* = 3–5) samples were still soluble, but the samples (*n* = 6–8) were precipitated and took AP-β sheet structures. On the other hand, the Gr(A)_n_Gr* (*n* = 4–8) samples were precipitated after adding methanol (MeOH) to the formic acid (FA) solutions.

Figure 8 shows the Ala Cβ peaks in the dipolar dephasing ^13^C CP / MAS NMR spectra of Gr(Ala)_n_Gr* (*n* = 4 and 6–8) after FA and MeOH treatment to change them to the AP-β sheet structure. Both the ① 22.7 ppm and ③ 19.8 ppm peaks are assigned to the AP-β sheet structure with a staggered packing arrangement and the ④ 16.8 ppm peak is assigned to random coil. The peak ② at 21.2 ppm is assigned to the general AP-β sheet structure rather than rectangular packing arrangement because the chemical shift shows the typical AP-β sheet structure and the poly-A sequence is not necessary to consider the possibility of the inter-molecular hydrogen bonding with bound waters. The MD simulation was performed for the sequence GGLGGQGAG(A)_6_GGAGQGGYGG without two (E)_4_ blocks to confirm the observed results as shown in Figure 9 [63].

Almost all Ala residues in the central sequence, (Ala)_6_, were in the AP-β sheet structure (96–98%); this finding was similar to the observed data for Gr(Ala)_6_Gr* (94.5%).

In addition, the staggered packing arrangement was preserved even after the annealing and replica exchange molecular dynamics (REMD) simulations, suggesting that staggered packing is a reasonably stable structure [63]. The ^13^C CP/MAS NMR spectra of a series of AP-β sheet poly-A samples were reported by Lee and Ramamoorthy [98] and Wildman et al. [99]. In their papers, a similar Ala Cβ spectral pattern-like staggered arrangement was reported, that is, an asymmetric peak at 19.9 ppm and a small peak at 22.9 ppm for AP-β sheet poly-A samples with high molecular weight. In general, X-ray diffraction is a very powerful technique used to study the packing structure, but such studies have been difficult to conduct because of the large amounts of amorphous fraction in dragline silk fiber as shown in Figure 7 for *N. clavata* dragline silk fibers [59].

## 5. Dynamical Behavior and Hydration of *N. clavipes* Dragline Silk

The unique characteristic of the spider silks, supercontraction, which occurs in the process of hydration [8,9,10], has been the interest of many NMR researchers studying the spider dragline silk [27,30,31,34,35,37,43,45,47,48,49,53,54,57,59,60,61]. For example, Figure 10 shows the aliphatic region from fully relaxed ^13^C directly detected MAS NMR spectrum of *N. clavipes* major ampullate silk in the hydrated state reported by Jenkins et al. [48]. Compared with Figure 2, it is noted that all peaks became shaper, i.e., three Ala Cβ, Glu Cβ and Glu Cγ, and two Ser Cβ peaks could be observed separately. Two Ala Cβ peaks at the AP-β sheet region were also observed as discussed in the previous section in detail.

Holland and Yarger groups studied the structure and dynamics of stable-isotope labeled spider dragline silks in the dry and hydrated states very actively using several advanced two-dimensional (2D) solid-state NMR, i.e., 2D ^13^C-^13^C correlation spectrum with dipolar assisted rotational resonance, 2D ^1^H-^13^C or 2D ^13^C-^15^N Heteronuclear correlation NMR, Homonuclear double quantum NMR and 2D incredible natural abundance double quantum transfer experiment as reviewed previously [89,90,92].

The structure and dynamics of *N. clavipes* dragline silks was also studied by using the selectively stable-isotope labeled 47-mer peptides flanked by (Ala)_6_ at both ends listed in Table 1 (II)[60]. Figure 11 is the expanded regions (10−70 ppm) of the ^13^C-labeled 47 mer peptides with three ^13^C-labeled amino acid motifs, (A) [2-^13^C]Gly^25^[3-^13^C]Ala^26^[1-^13^C]Gly^27^ or (B) [2-^13^C]Gly^35^[3-^13^C]Ala^36^[1-^13^C]Gly^37^, respectively in the dry and hydrated states [60]. Here, (a) ^13^C refocused insensitive nuclei enhanced by polarization transfer (r-INEPT) [100], (b) ^13^C dipolar decoupled magic angle spinning (DD/MAS) [101,102], and (c) ^13^C CP/MAS NMR spectra in the hydrated states, and (d) ^13^C CP/MAS spectrum in the dry state. The ^13^C r-INEPT, where the pulse sequence was developed for solution NMR, is sensitive to the mobile component of the peptide with fast isotropic motion (>10^5^ Hz) [103]. In contrast, ^13^C CP/MAS NMR can observe selectively immobile components or those with very slow motion (<10^4^ Hz) [103]. If the penetration of water molecules causes an increase in the chain mobility, a loss in the CP signal of the amino acid residues occurs and consequently such a mobile domain cannot be observed in the ^13^C CP/MAS NMR spectra [43]. ^13^C DD/MAS NMR can be used to detect the mobile domains as well as the immobile domains [102]. Thus, these three kinds of ^13^C NMR techniques, ^13^C r-INEPT, ^13^C CP/MAS, and ^13^C DD/MAS NMR, provide different perspectives on the dynamic behavior of hydrated peptides and can be used together to characterize their local structure and conformations [57,59,60,104,105]. The ^13^C CP/MAS NMR spectra of two peptides in the dry state are the same as those in Figure 5. With hydration, sharp peaks in the Gly Cα and Ala Cβ carbons at 43.0 and 16.6 ppm, respectively assigned to the random coil conformation with high mobilities can be observed in the ^13^C CP/MAS NMR spectrum (c) of the peptide (A). Additionally, these sharp peaks became dominant in the ^13^C DD/MAS NMR (b) and ^13^C r-INEPT (a) spectra.

Similar changes in the spectra were also observed for another 47-mer peptide with three ^13^C-labeled amino acid motifs, [2- ^13^C]Gly^12^ [3-^13^C]Ala^13^[1-^13^C]Gly^14^ (not shown). In contrast to these observations, the effect of the hydration of the peptide (B), [2-^13^C]Gly^35^ [3-^13^C]Ala^36^ [1-^13^C]Gly^37^-47 mer peptide was quite different. Only a small difference in the ^13^C CP/MAS NMR spectra was observed between the dry (d) and hydrated states (c). In addition, the ^13^C DD/MAS NMR and ^13^C r-INEPT spectra ((b) and (a), respectively) showed very small sharp peaks with random coil chemical shifts in the hydrated state. Thus, the effect of hydration is very small for the motif, [2-^13^C]Gly^35^ [3-^13^C]Ala^36^ [1-^13^C]Gly^37^ because the motif is adjacent to the (Ala)_6_(Glu)_4_ sequences with AP-β structure. These results indicate that the hydration occurs heterogeneously along the Gly-rich region. The MD simulations are useful in examining changes in the local conformations of the individual residues of silks in the dry and hydrated states from the theoretical perspective [42,52,62,68,76,77,78,79,80,81,82]. Figure 12 shows the distributions of the Φ-Ψ torsion angles of the ^13^C labeled residues in the motifs, Gly^12^Ala^13^Gly^14^, Gly^25^Ala^26^Gly^27^, and Gly^35^Ala^36^Gly^37^ of the Gly-rich region in the dry and hydrated states, respectively, as calculated by MD simulation [60]. As is expected, the distributions of Φ-Ψ torsion angles of the residues become generally larger due to partial breaking of the hydrogen bonds in peptides by water molecules. Changes in the Φ-Ψ torsion angles of the motif, Gly^25^Ala^26^Gly^27^, shown in Figure 12 follow such a prediction. Thus, broader distributions of Φ-Ψ torsion angles by hydration indicate the occurrence of fast exchange among several conformations in water. This supports the appearance of the sharp peaks in the ^13^C DD/MAS and ^13^C r-INEPT NMR spectra as shown in Figure 11A. On the other hand, this is not the case for the motif, Gly^35^Ala^36^Gly^37^, where with hydration the change in the Φ-Ψ torsion angles was small, especially for the residues, Ala^36^Gly^37^, and the distribution of the Φ-Ψ torsion angles was confined to the AP-β sheet region. Thus, the residues close the C-terminal poly-Ala sequence, the Gly^37^ and Ala^36^ residues, were structured and stable. This is in agreement with the experimental results (Figure 11B).

## 6. Dynamical Behavior and Hydration of *N. clavata* Dragline Silk

*N. clavata* is often known as the Joro spider in East Asia (Japan, China, Korea, and Taiwan) and can be easily obtained in Japan [106,107]. Therefore, the structural and dynamical studies of dragline silk were performed for *N. clavata*. Figure 13 is the expanded regions (10−70 ppm) of four kinds of ^13^C sold-state NMR spectra of *N. clavata* dragline silk fibers in the dry and hydrated states, i.e., (a) ^13^C r-INEPT, (b) ^13^C DD/MAS, and (c) ^13^C CP/MAS NMR spectra in the hydrated states, and (d) ^13^C CP/MAS spectrum in the dry state [59]. The intensities of the ^13^C CP/MAS NMR peaks except for Ala Cα and Cβ peaks decreased significantly in the hydrated state (c), which was due to a significant loss in CP signals of the carbons whose mobility increased by hydration as mentioned above. Actually, several small peaks for the other amino acid residues in the ^13^C CP/MAS NMR spectra in the hydrated state (c) were observed as peaks with increased intensities in the ^13^C DD/MAS NMR spectrum (b) and as sharp peaks in the ^13^C r-INEPT spectrum (a) in the hydrated state. Only the random coil peak of the Ala residue was observed in the ^13^C r-INEPT spectrum, and the Ala peaks in the poly-A sequences with AP-β sheet structure were no longer observed.

This meant that water molecules could penetrate inside the random coil domains of *N. clavata* dragline silk fibers, but hardly penetrated through the hydrophobic β-sheet domains of poly-A sequences. The Ala Cβ peak region was analyzed in detail for [3-^13^C]Ala- *N. clavata* dragline silk fibers. It is speculated that there are also contributions from the peaks of the Leu and Arg side chains, especially at the lower field region, 20-24 ppm in the chemical shift range from 14 to 24 ppm of the Ala Cβ peak in the ^13^C natural abundant spectrum of *N. clavata* dragline silk as shown in Figure 13. However, these additional contributions to the Ala Cβ peak can be neglected by [3-^13^C]Ala labeling, which makes it possible to analyze these Cβ spectral region quantitatively.

Figure 14 shows Ala Cβ peaks in the ^13^C CP/MAS NMR spectra of [3-^13^C]Ala-*N. clavata* dragline silk fiber in the dry and hydrated states [59]. The percentages of random coil changed from 23% in the dry state to 10% in the hydrated state in the ^13^C CP/MAS NMR spectra (A). As mentioned above, a significant loss in CP signals occurred for the Ala Cβ carbons with random coil because of increased mobility by penetration of water molecules in the hydrated state. In addition, the line shape became sharper in the hydrated state. From the Ala Cβ peaks (B) with exclusively AP-β sheet structures, the percentage of the general AP-β sheet and staggered packing arrangement were determined as 49% and 51%, respectively in the dry state, and 40% and 60%, respectively in the hydrated state.

Solid-state NMR has the advantage that it can provide detailed structural information and can also give the dynamical information at different motional levels from the side-chain fast motion to the slower motion of the backbone chain of the spider dragline silks.

Thus, the relaxation behavior of Ala residues in [3-^13^C]Ala-*N. clavata* dragline silk fiber was studied in relation to the difference in the packing structures in the dry and hydrated states [61]. Figure 15 shows a series of partially relaxed ^13^C CP/MAS NMR spectra of Ala Cβ peaks of [3-^13^C]Ala-*N. clavata* dragline silk fiber in the dry state (A) and in the hydrated state (B) as a function of delay time τ for T_1_ determinations at 25 °C. The T_1_ values were 0.61 ± 0.02 s for 16.6 ppm (random coil), 0.57 ± 0.02 s for AP-β sheet (main peak at 20.2 ppm: mixture of rectangular and staggered peaks), and 0.93 ± 0.02 s for AP-β sheet (22.9 ppm: lower field staggered peak). On the other hand, the T_1_ values in the hydrated state were 0.48 ± 0.01 s for 17.2 ppm peak, 0.44 ± 0.02 s for 20.2 ppm peak, and 0.88 ± 0.02 s for 22.9 ppm, which are slightly shorter than those of the corresponding peaks in the dry state. Thus, it is noted that the T_1_ value of the 22.9 ppm was significantly longer than other two peaks in both dry and hydrated states. The longer T_1_ value means a faster motion, which was obtained from the temperature dependent T_1_ experiment [61]. Thus, the motional behavior of the lowest field peak assigned exclusively staggered packing structure shows the fastest hopping motion of the Cβ carbon about the C_3_ axis of Ala Cβ group in the dry and hydrated states [108,109].

The origin of these long T_1_ values in the staggered packing structure was already studied from the ^13^C solid-state NMR relaxation work of *S. c. ricini* silk fibroin fiber as shown in Figure 16 [110,111,112,113]. Especially, the MD simulation was performed to interpret the relationship between the packing structure and the dynamical behavior of these Ala Cβ groups in the poly-A region of *S. c. ricini* silk fibers [110]. Namely, two of the Ala Cβ carbons out of eight existing in the unit cell of the staggered packing structure exhibited the fastest hopping motion in spite of the shortest Cβ−Cβ distance. The unusual motion of the Ala methyl groups is named as a geared hopping motion, as shown in Figure 16.

## 7. Conclusions and Future Aspects

In this review, the author showed how solid-state NMR coupled with stable isotope labeling of the silks and the related silk peptides, and combination of solid-state NMR and MD simulation have contributed to a better understanding the structure and dynamics of the spider dragline silks in the dry and hydrated states. Recently, the excellent mechanical properties of spider silk fiber have been challenged to predict on the basis of the atomic resolution structure of the crystalline region embedded in a softer-amorphous region using MD simulations [73,74,75,76,77,78,79,80,81,82,83,84,85]. In the future, such a prediction of the mechanical properties of natural and recombinant spider silks [11,57,59,61] is expected to increase more and more for molecular design of new materials with excellent mechanical properties. Therefore, a more detailed investigation about the atomic resolution of spider silk fiber seems to be required experimentally. Moreover, it is important to accumulate a lot of knowledge about the structures and dynamics of other silks and/or silks of other species, including smaller orb weavers and non-orb weavers, and also change in the structure and dynamics of the silks to explain spider silk property variations in ecological and evolutionary contexts other than hydration [114]. NMR spectroscopy, especially solid-state NMR including dynamic nuclear polarization (DNP) [115] will undoubtedly use more frequently for the purpose.

## Figures and Tables

**Figure 1 molecules-25-02634-f001:**
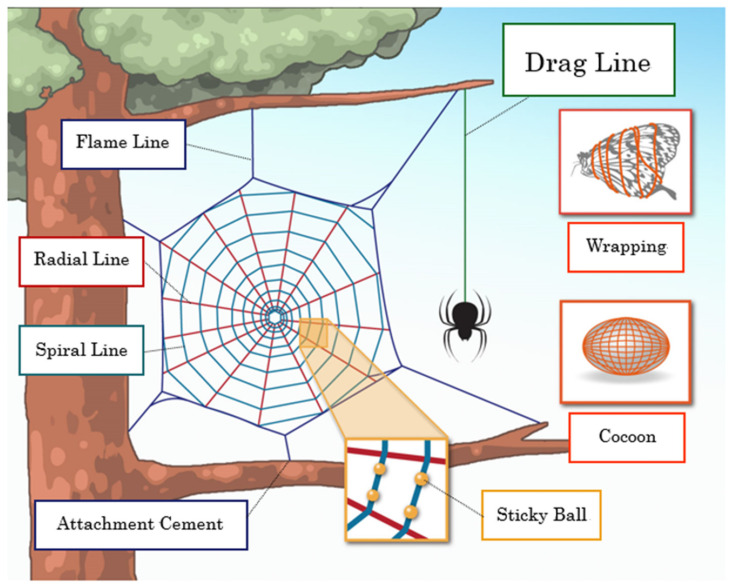
Schematic overview of seven types of spider silks.

**Figure 2 molecules-25-02634-f002:**
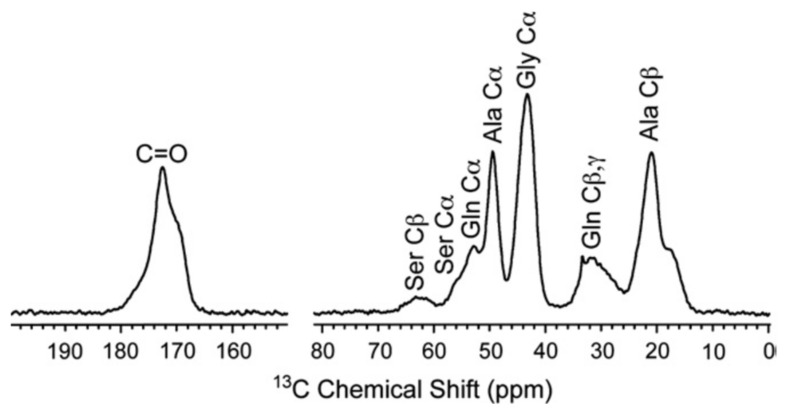
^13^C CP/MAS NMR spectrum of dragline silk fiber from *N. clavipes* spider together with the assignment [26]. The chemical shifts are represented from tetramethylsilane (TMS).

**Figure 3 molecules-25-02634-f003:**
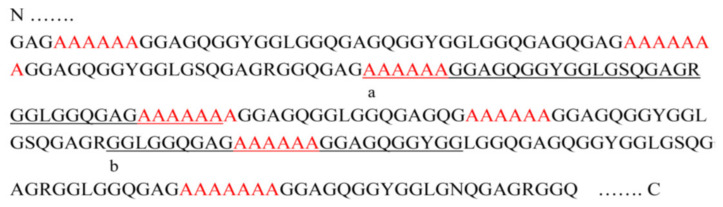
Partial amino acid sequence of MaSp1 spider silk protein. Polyalanine sequences (red) forms antiparallel β-sheet structure [4]. The underlined sequence, a, is the selected sequence to study the conformations of amino acid residues in the Gly-rich region sandwiched by poly-A sequence on both sides reported previously [46,60,62]. The underlined sequence, b, is selected to study the conformation and packing structure of poly-A sequence sandwiched by partial Gly-rich sequences [63].

**Figure 4 molecules-25-02634-f004:**
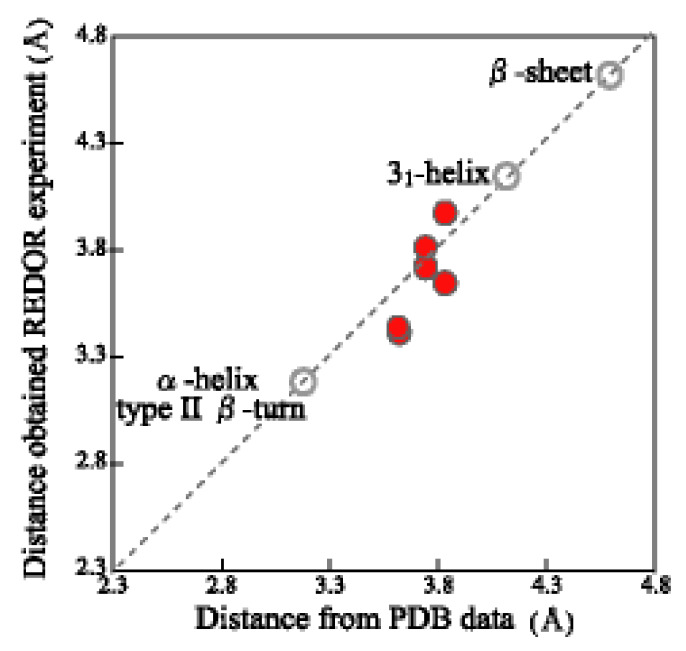
Plot of the REDOR-determined inter-nuclear distances of the ^13^C–^15^N nuclei in Table 1(I) vs. the corresponding distances calculated from the PDB database. The red circles indicate the inter-nuclear distances between [1-^13^C]Gly and [^15^N]Gly nuclei in the [1-^13^C]Gly-X-[^15^N]Gly (X = Q,Y,L, and R) motifs determined by REDOR experiment in Table 1(I) and those calculated from the PDB database. The grey hollow circles are corresponding distances of typical conformations: anti-parallel β-sheet; 4.6 Å, α-helix and type II β-turn; 3.2 Å and 3_1_-helix; 4.2 Å [46].

**Figure 5 molecules-25-02634-f005:**
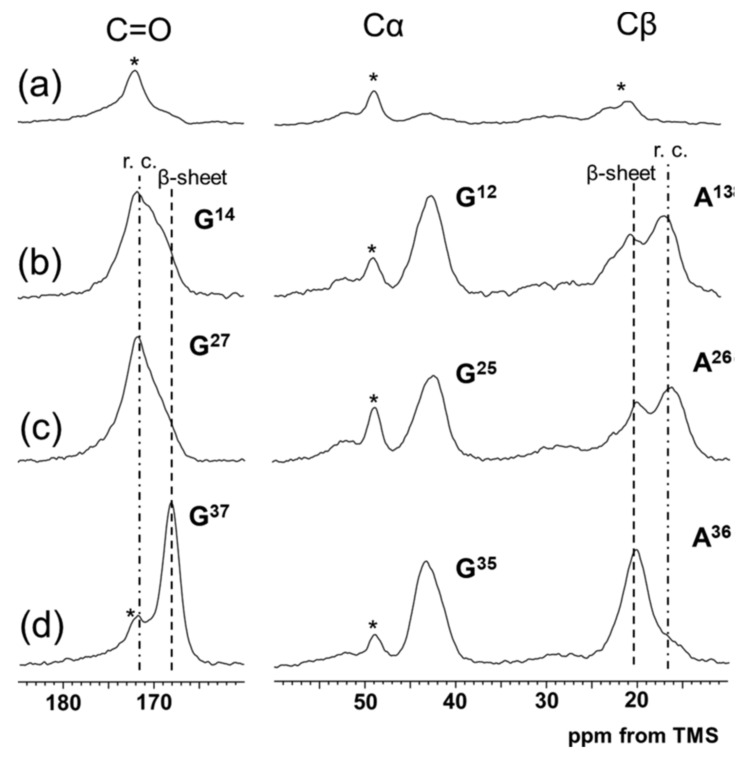
^13^C CP/MAS NMR spectra of ^13^C successively triple-labeled 47-mer peptides incorporated as the [2-^13^C]Gly [3-^13^C]Ala[1-^13^C]Gly motif in different positions (Table 1 (II)): (**a**) non-labeled 47-mer peptides (Sample 1), (**b**) [2-^13^C]Gly^12^[3-^13^C]Ala^13^ [1-^13^C]Gly^14^ motif (Sample 8), (**c**) [2-^13^C]Gly^25^[3-^13^C]Ala^26^[1-^13^C]Gly^27^ motif (Sample 9), and (**d**) [2-^13^C]Gly^35^[3-^13^C]Ala^36^ [1-^13^C]Gly^37^ motif (Sample 10) [62]. The asterisk,* shows one of the natural abundance Ala Cβ, Cα, and C = O peaks of two (Ala)_6_ sequences in the 47-mer peptides.

**Figure 6 molecules-25-02634-f006:**
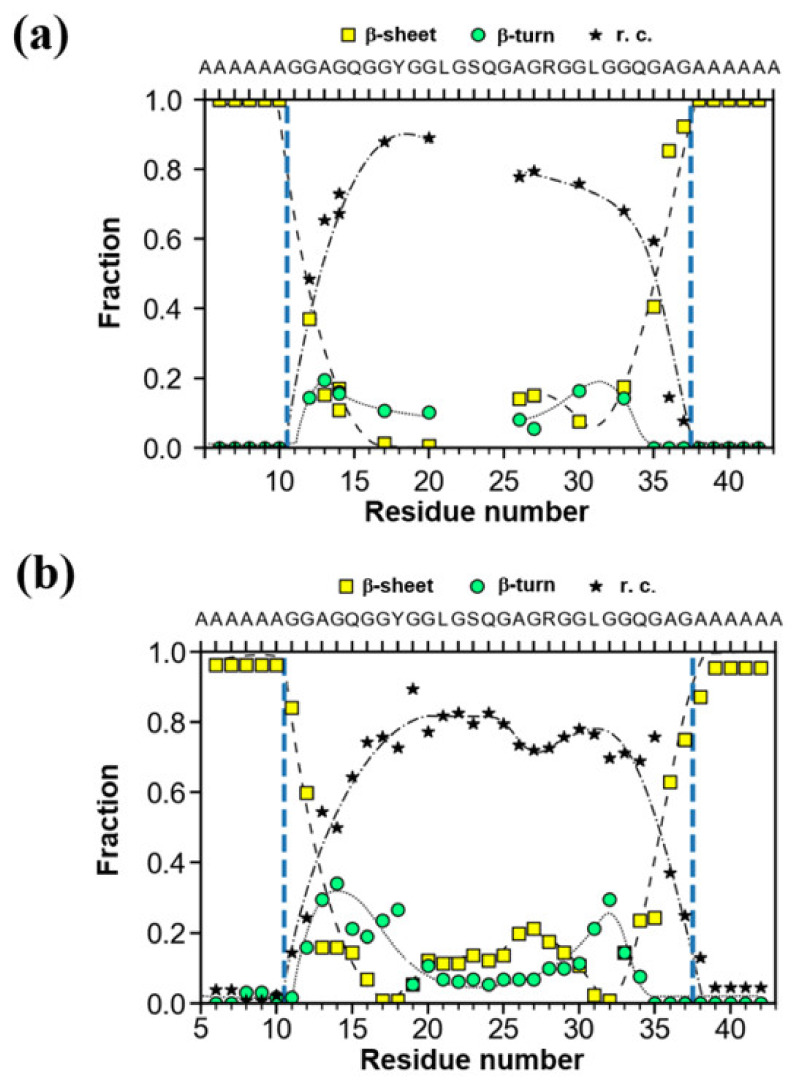
(**a**) The fractions of several conformations, i.e., AP-β sheet, random coil and β-turn determined using ^13^C CP/MAS NMR for the ^13^C-labeled individual residues in 47-mer peptide, (E)_6_(A)_6_GGAGQGGYGGLGSQGAGRGGLGGQGAG(A)_6_(E)_6_ (Table 1 (I)) as the sequential model of the Gly-rich region (Figure 3a). The broken, dash-dotted and solid lines indicate change in the fractions of AP-β sheet, random coil and β-turn conformations, respectively as a function of residue number. (**b**) The fractions of several conformations calculated by the structural identification (STRIDE) algorithm, for the representative structure of the replica exchange molecular dynamics (REMD) simulations with the lowest temperature replica for the individual residues in the sequence, (A)_6_GGAGQGGYGGLGSQGAGRGGLGGQGAG(A)_6_ shown for a comparison [62]. The arrangement of the peptide chains with the sequences before MD simulation was as follows. The four extended peptide chains were oriented in antiparallel forms within the planes and parallel between the planes. Three parallel planes were assumed. The intermolecular hydrogen bonding systems of the (A)_6_ blocks were set in a staggered packing arrangement.

**Figure 7 molecules-25-02634-f007:**
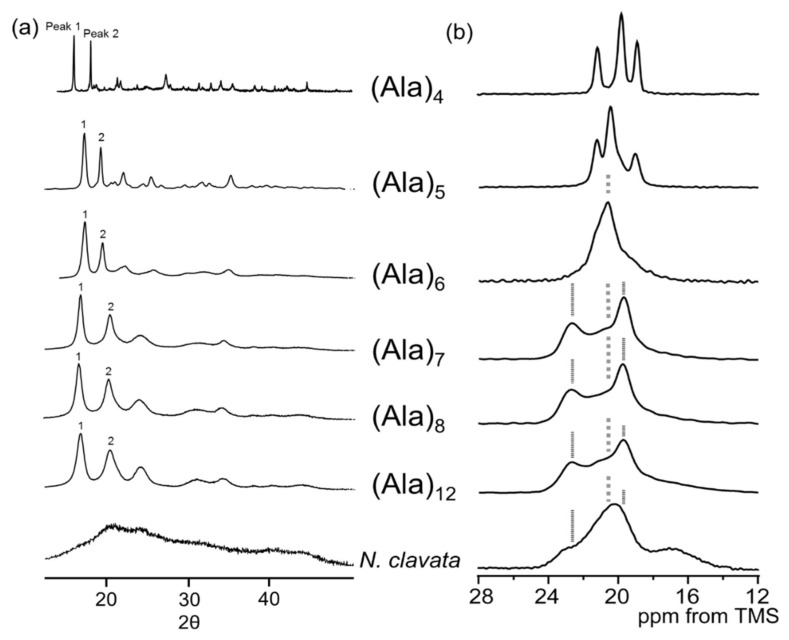
(**a**) X-ray diffraction patterns and (**b**) Ala Cβ peaks in the ^13^C CP/MAS NMR spectra of alanine oligopeptides, (Ala)_n_ (n = 4−8 and 12), together with those of *N. clavata* dragline silk fibers [52].

**Figure 8 molecules-25-02634-f008:**
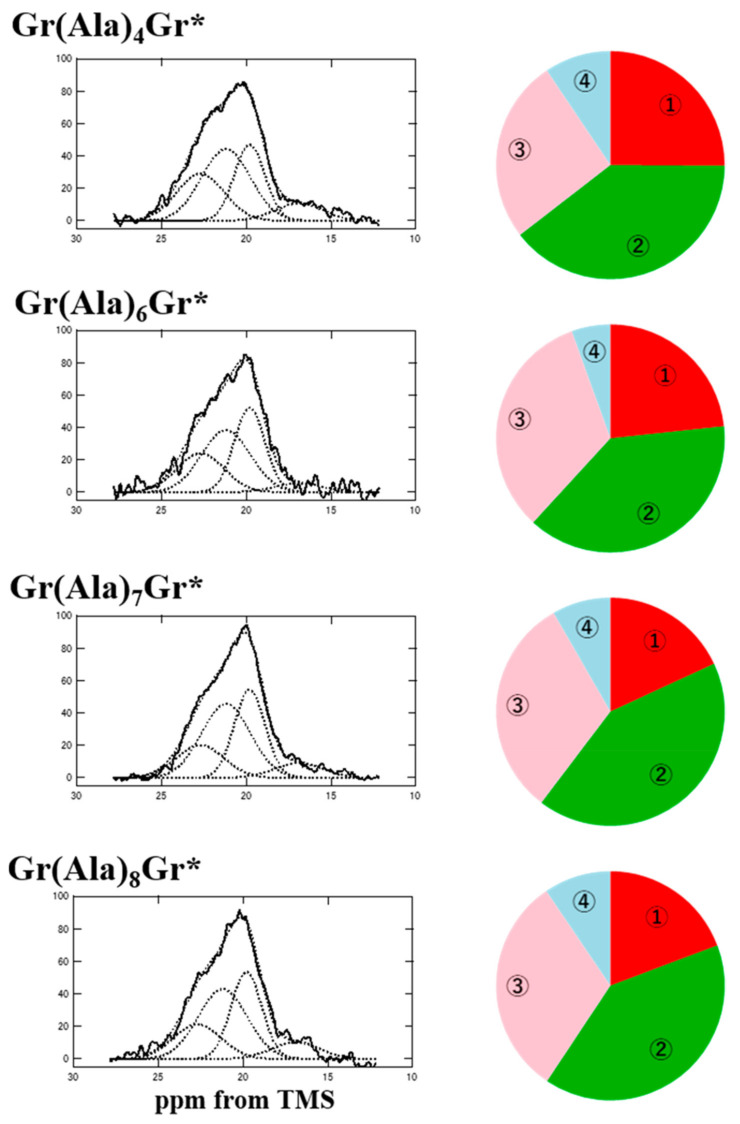
(**Left**) Expanded dipolar dephasing ^13^C DD/MAS NMR spectra of the Ala Cβ regions of Gr(Ala)_n_Gr* (*n* = 4 and 6−8) after FA/MeOH treatment [63]. The decomposed four spectra shown by broken lines were obtained by the deconvolutions under assuming the chemical shifts as 22.7, 21.2, 19.8 and 16.9 ppm from lower field to higher field, respectively. (**Right**) Pie charts of the fractions determined by the deconvolutions. The color of the fraction is ① red (22.7 ppm), ② green (21.2 ppm), ③ pink (19.8 ppm), and ④ light blue (16.9 ppm; right).

**Figure 9 molecules-25-02634-f009:**
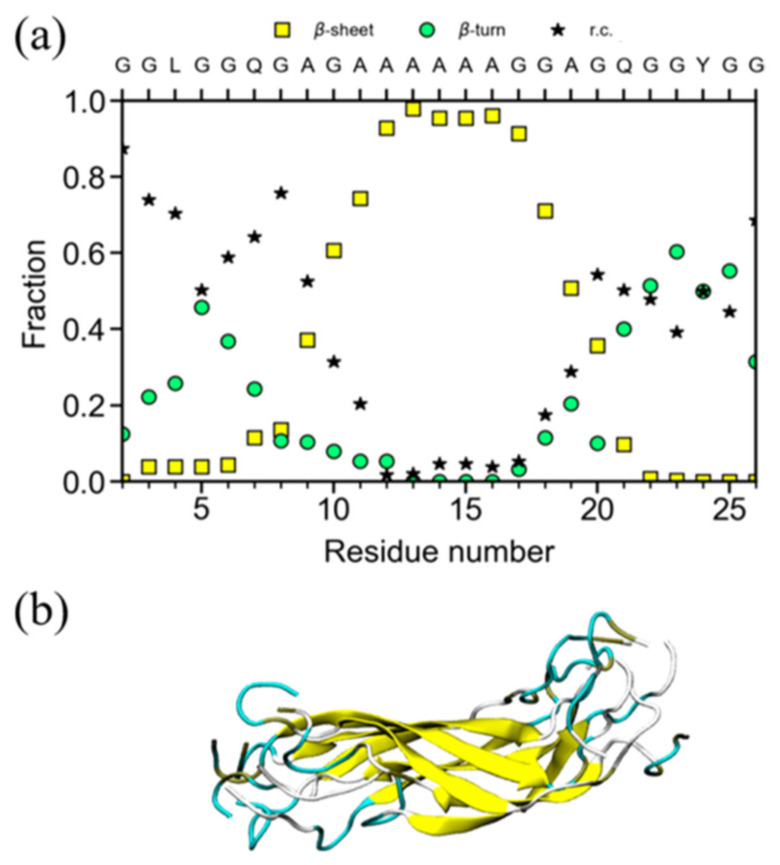
(**a**) Plot of the fractions determined by the STRIDE algorithm, for the representative structure of the REMD simulation with the lowest temperature replica of the sequence, GGLGGQGAG(A)_6_G- GAGQGGYGG. (**b**) Image of the backbone structure after the MD simulation [63].

**Figure 10 molecules-25-02634-f010:**
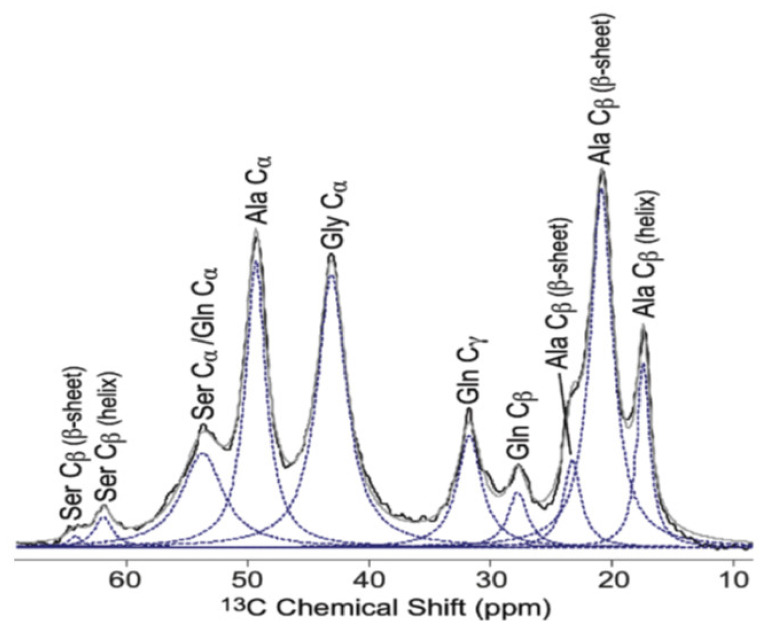
Expanded aliphatic region from fully relaxed ^13^C directly detected MAS NMR spectrum of *N. clavipes* major ampullate silk in the hydrated state. The fraction of each peak was determined from the deconvolution shown by the broken lines [48].

**Figure 11 molecules-25-02634-f011:**
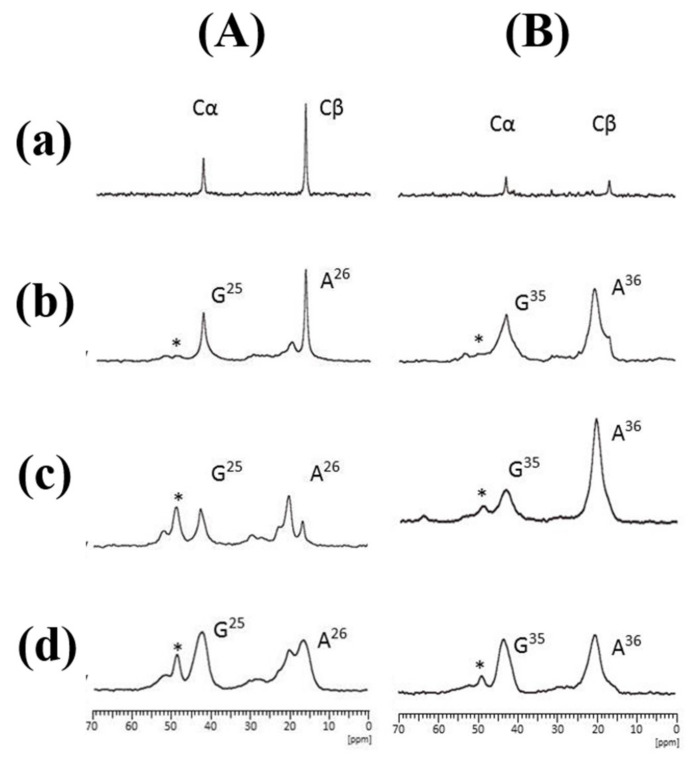
^13^C solid-state NMR spectra (**a**–**d**) of 47-mer peptides with ^13^C-labeled motifs; (**A**) [2-^13^C]Gly^25^[3-^13^C]Ala^26^ [1-^13^C]Gly^27^ and (**B**) [2-^13^C]Gly^35^[3-^13^C]Ala^36^ [1-^13^C]Gly^37^. (**a**) ^13^C r-INEPT, (**b**) ^13^C DD/MAS and (**c**) ^13^C CP/MAS spectra in the hydrated state, and (**d**) ^13^C CP/MAS spectrum in the dry state [60]. The peaks marked by asterisk * were assigned to Ala Cα peaks of non-labeled (Ala)_6_ in the peptides.

**Figure 12 molecules-25-02634-f012:**
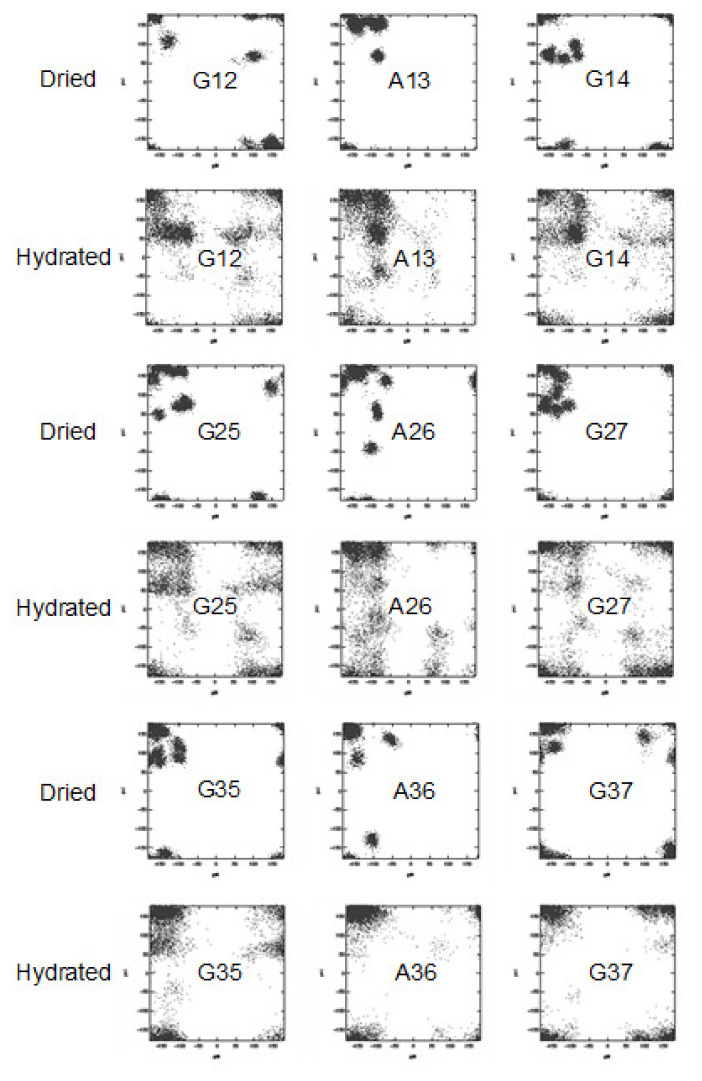
Ramachandran maps showing the distributions of (Φ,Ψ) torsion angles of the residues in the motifs, Gly^12^Ala^13^Gly^14^, Gly^25^Ala^26^Gly^27^, and Gly^35^Ala^36^Gly^37^ of the Gly-rich region in the dry and hydrated states, respectively, as calculated by MD simulation [60].

**Figure 13 molecules-25-02634-f013:**
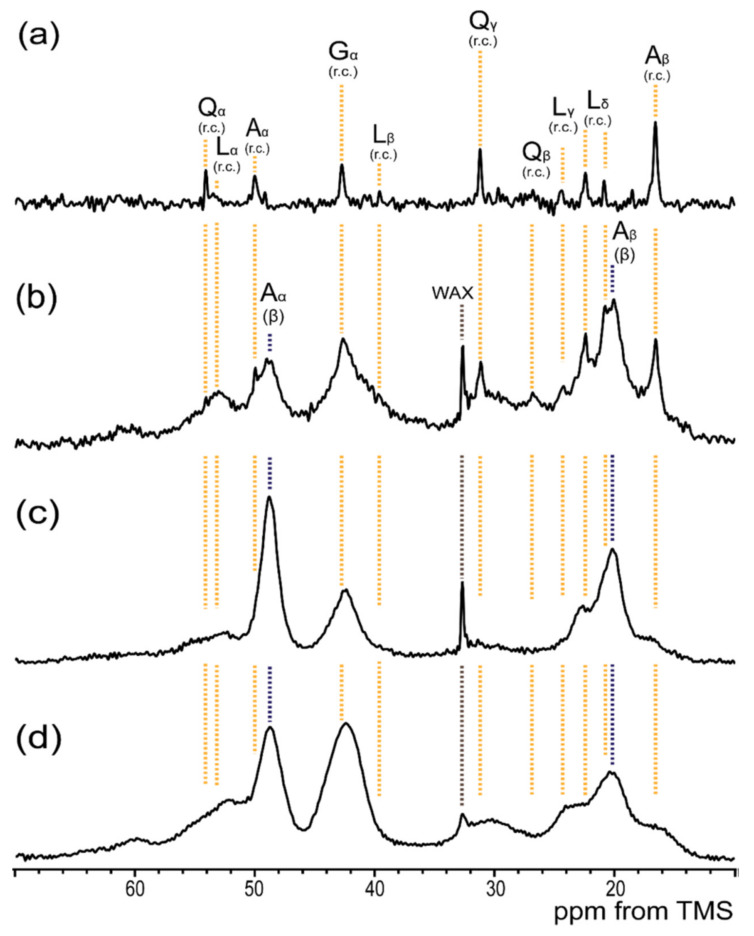
Expanded ^13^C solid-state NMR spectra (10−70 ppm) of *N. clavata* dragline silk fibers together with assignments: (**a**) ^13^C refocused INEPT, (**b**) ^13^C DD/MAS, and (**c**) ^13^C CP/MAS spectra in the hydrated state and (**d**) ^13^C CP/MAS spectrum in the dry state [59].

**Figure 14 molecules-25-02634-f014:**
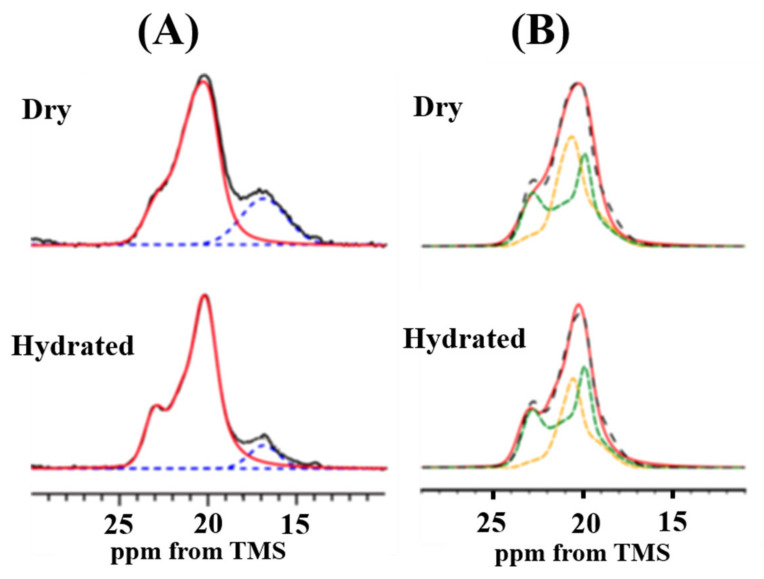
(**A**) Expanded Ala Cβ peaks in the ^13^C CP/MAS spectra (10−30 ppm) of [3-^13^C]Ala-*N. clavata* dragline silk fibers in the dry and hydrated states together with peak deconvolution [59]. (**A**) The observed spectra (black solid lines) were deconvoluted by random coil (blue broken lines) and β-sheet structure (red solid lines). (**B**) The expanded Ala Cβ peaks with exclusively β-sheet structures were reproduced by superpositions (black broken lines) of the Ala Cβ peak patterns of (Ala)_6_ (orange broken lines) and (Ala)_7_ (green broken lines).

**Figure 15 molecules-25-02634-f015:**
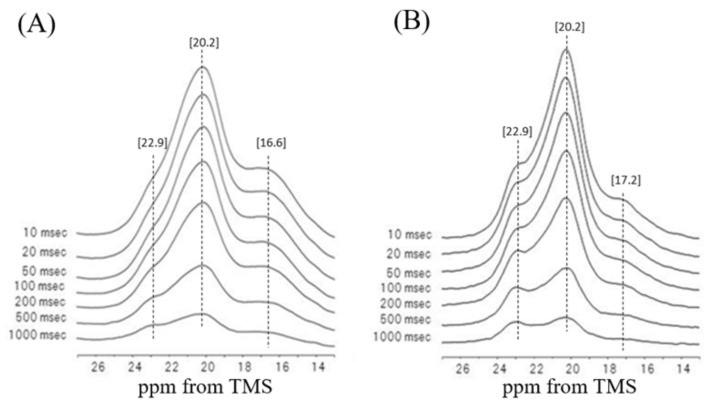
A series of partially relaxed ^13^C CP/MAS NMR spectra of Ala Cβ peaks of [3-^13^C]Ala-*N. clavata* dragline silk fiber in the dry state (**A**) and in the hydrated state (**B**) [61].

**Figure 16 molecules-25-02634-f016:**
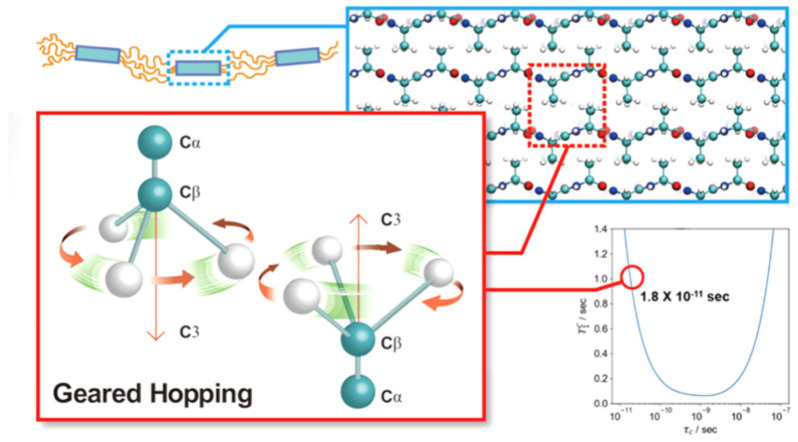
The packing structure and the dynamical behavior of these Ala Cβ groups in the poly-A region of *S. c. ricini* silk fibers studied by the MD simulation [110]. Two of the Ala Cβ carbons out of eight existing in the unit cell of the staggered packing structure exhibited the fastest hopping motion in spite of the shortest Cβ−Cβ distance, indicating a geared hopping motion [110].

**Table 1 molecules-25-02634-t001:** 47-mer peptides as the sequential models of the Gly-rich region in Figure 3 as underlined sequence, a flanked by (Ala)_6_ at both ends [46,60,62]. (I) The peptides, 2–7 were used for determinations of both the inter-nuclear distances of ^15^N and ^13^C labeled sites, and the fractions of the local conformations of ^13^C labeled Gly residues. (II) The peptides, 8–10 were used for determination of the fractions of the local conformations of the ^13^C labeled Gly-Ala-Gly motifs.

I:	REDOR and ^13^C CP/MAS NMR Experiments	Distance *
1	(E)_4_(A)_6_GGAGQGGYGGLGSQGAGRGGLGGQGAG(A)_6_(E)_4_	
2	(E)_4_(A)_6_GGA**[1-^13^C]G^14^**Q**[^15^N]G^15^**GYGGLGSQGAGRGGLGGQGAG(A)_6_(E)_4_	3.71 Å
3	(E)_4_(A)_6_GGAGQG**[1-^13^C]G^17^**Y**[^15^N]G^19^**GLGSQGAGRGGLGGQGAG(A)_6_(E)_4_	3.97 Å
4	(E)_4_(A)_6_GGAGQGGYG**[1-^13^C]G^20^**L**[^15^N]G^22^**SQGAGRGGLGGQGAG(A)_6_(E)_4_	3.41 Å
5	(E)_4_(A)_6_GGAGQGGYGGLGSQGA**[1-^13^C]G^27^**R**[^15^N]G^29^**GLGGQGAG(A)_6_(E)_4_	3.64 Å
6	(E)_4_(A)_6_GGAGQGGYGGLGSQGAGRG**[1-^13^C]G^30^**L**[^15^N]G^32^**GQGAG(A)_6_(E)_4_	3.43 Å
7	(E)_4_(A)_6_GGAGQGGYGGLGSQGAGRGGLG**[1-^13^C]G^33^**Q**[^15^N]^35^**GAG(A)_6_(E)_4_	3.81 Å
**II:**	**^13^** **C CP/MAS NMR Experiment**	
8	(E)_4_(A)_6_G **[2-^13^C]G^12^ [3-^13^C]A^13^ [1-^13^C]G^14^**QGGYGGLGSQGAGRGGLGGQGAG(A)_6_(E)_4_	
9	(E)_4_(A)_6_GGAGQGGYGGLGSQ **[2-^13^C]G^25^ [3-^13^C]A^26^ [1-^13^C]G^27^**RGGLGGQGAG(A)_6_(E)_4_	
10	(E)_4_(A)_6_GGAGQGGYGGLGSQGAGRGGLGGQ **[2-^13^C]G^35^ [3-^13^C]A^36^ [1-^13^C]G^37^**(A)_6_(E)_4_	

Distance * is the inter-nuclear distance between ^15^N and ^13^C labeled sites [46].

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
