# Peer review of "Structure and Dynamics of Spider Silk Studied with Solid-State Nuclear Magnetic Resonance and Molecular Dynamics Simulation"

_molecules, 2020, doi:10.3390/molecules25112634_

Round 1
Reviewer 1 Report
This contribution is a review of recent contributions using combined solid-state NMR, especially via selective isotopic enrichment, and molecular dynamics calculations to understand the local atomic scale and dynamics of silks, written by a key contributor to this field. The review is authoritative and well put together concentrating recent developments. The range of modern NMR spectroscopy approaches that provide sensitivity to motion, along with direct relaxation time measurements and cross referencing this to the computational work via the molecular dynamics is interesting. The examples illustrate very nicely some of the key insights that this approach can give to this intriguing class of materials. This will be a very helpful review that gives a very up to date overview of the literature, fitting in with the special issue well. There is a good selection of illustrative examples from two spider silks, as well as concentrating on the structural and dynamical effects of hydration. The paper can certainly be published. It could do with a good presentational edit as in places minor errors and unusual sentence constructions make it more difficult to read than it could be. There is one minor scientific issue commented on below. If these minor points are examined and addressed, publications in fully recommended.
Minor scientific point
- p16, lines 365-368, several T1 values are mentioned. These are experimentally determined, so some indication of the accuracy/associated error should be quoted.
Minor typographical/presentational corrections
- p1, line 3, there is only one author, so should be author’s
- p1, lines 21-22, sentence ‘In addition…tools’ needs rewriting
- p1, line 30 delete a before unique
- p1, line 32, possess mis-spelt
- p1, line 33, delete the . after 1
- p1, line 39, delete been
- p1, line 40, insert a space before and
- p2, line 44, delete s to give provide
- p2, Figure 1, attachment on the figure is incorrectly spelt
- p2, line 73, add s to give hopes
- p3, lime 80, spinneret is mis-spelt
- p4, line 122, insert a space after Fig.
- p5, line 143 the reference should be [33]
- p8, line 197, the symbols seen to have been corrupted
- p8, in Figure 7, the units on the scale at the bottom have been cut off
- p11, line 255, delete the space after two-
- p11, sentences finish mid-line and should be tidied up, for example run line 256 directly on to 260. Also lines 260-262 this text should not be bold.
- p11, line 268, insert , after [100]
- p13, line 296, after Gly37 there needs to be a space
- p13, lines, 302, 304 and 305 the symbol has been corrupted
- p15, line 335, insert a space after 24
- p15, line 337, add an s to contribution
- p16, line 353, replace there was with a
Author Response
Thank you for a high evaluation about this review. The followings are the answer to the review of reviewer 1.
Minor scientific point
1. p16, lines 365-368, several T1 values are mentioned. These are experimentally determined, so some indication of the accuracy/associated error should be quoted.
Answer
The experimental error in the T1 determination for each peak was included in the revised text as follows. The T1 values were 0.61±0.02 s for 16.6 ppm (random coil), 0.57±0.02 s for β-sheet (main peak at 20.2 ppm: mixture of rectangular and staggered peaks), and 0.93±0.02 s for β-sheet (22.9 ppm: lower field staggered peak). On the other hand, the T1 values in the hydrated state, were 0.48±0.01 s for 17.2 ppm peak, 0.44±0.02 s for 20.2 ppm peak and 0.88±0.02 s for 22.9 ppm which are slightly shorter than those of the corresponding peaks in the dry state.
Minor typographical/presentational corrections
Answer
According to reviewer’s corrections, we revised them. Thank you so much for careful corrections which are very helpful for me.
2. p1, line 3, there is only one author, so should be author’s
3. p1, lines 21-22, sentence ‘In addition...tools’ needs rewriting
4. p1, line 30 delete a before unique
5. p1, line 32, possess mis-spelt
6. p1, line 33, delete the . after 1
7. p1, line 39, delete been
8. p1, line 40, insert a space before and
9. p2, line 44, delete s to give provide
10. p2, Figure 1, attachment on the figure is incorrectly spel
11. p2, line 73, add s to give hopes
12. p3, lime 80, spinneret is mis-spelt
13. p4, line 122, insert a space after Fig.
14. p5, line 143 the reference should be [33]
15. p8, line 197, the symbols seen to have been corrupted
16. p8, in Figure 7, the units on the scale at the bottom have been cut off
17. p11, line 255, delete the space after two-
18. p11, sentences finish mid-line and should be tidied up, for example run line 256
directly on to 260. Also lines 260-262 this text should not be bold.
19. p11, line 268, insert , after [100]
20. p13, line 296, after Gly37 there needs to be a space
21. p13, lines, 302, 304 and 305 the symbol has been corrupted 22. p15, line 335, insert a space after 24
23. p15, line 337, add an s to contribution
24. p16, line 353, replace there was with a
Reviewer 2 Report
Review of “Structure and Dynamics of Spider Silk…” by Asakura, T.
This paper reviews the literature showing the effectiveness of solid-state NMR methods for elucidating the structures of spider silk, a rapidly growing field with some recent exciting discoveries, including many by the author.
I didn’t have time to check the outcomes cited against the literature but assume the author, as an authority on the subject, to have done due diligence. I nevertheless do suggest the author check cited datasets and their interpretations for accuracy.
I have identified nonetheless some broader issues that I request the author address regarding assumptions about spider silk diversity, silk properties, structures, and variability thereof.
Firstly, the author concentrates the review almost exclusively on major ampullate silk. This is because it is the silk for which the vast majority of ssNMR has been done for. This is probably because of the volumes needed for ssNMR are only attainable with MA silk, and even then for particularly large orb weaving spiders such as Nephila spp. This is fine, but the author should acknowledge this and state that little is known of the other silks. However, as the techniques evolve we may be able to work with other spider silks and/or silks of other species, including smaller orb weavers and non-orb weavers, e.g. RTA clade spiders, cribellate spiders, Mygalamorphs.
On page 2 the authors compared the use of ssNMR with that of SAXS/WAXS and other spectroscopic techniques and concluded NMR to be superior. This is not necessarily true. ssNMR can certainly do things that SAXS/WAXS etc. cannot (e.g. on the fly analyses of amino acid distribution and the relative contributions of alanine and glycine in key secondary structures), but SAXS/WAXS and so on can do other things that NMR cannot (e.g. visualization of the orientation and alignment across the crystalline and amorphous regions). I think a more even-handed comparison is required, showing, where applicable, where NMR has added value to other types of analyses. Jeff Yarger’s work has many examples, which the author would know.
The author has assumed all spider silk to be represented by Nephila clavipes/clavata MA silks. That there are other silks that differ from MA silks I commented on above (so addressing that comment will suffice). However even MA silk is not homogeneous in structure and function. A single species, even individual, spider’s MA silk has a high degree of variability across spinning conditions (free fall vs forces vs anesthetized/unaethetized), diets (Blamires’, C. Craig, Tso’s papers), geography (Blamires/Tso), microhabitat, and exposure to toxins (e.g. insecticides) etc (reviewed by Blamires et al. 2017 Ann Rev Entomol). To date, no good NMR studies have been done to assess this variability at a nano-scale (although some good SAXS/WAXS work has been done), but the potential exists for studies to be done, for instance by using DNP techniques to smooth out the noise. I encourage the author to acknowledge the types and causes of variability and touch on the possibilities for assessing it using NMR.
I have noticed by tracking the literature cited within the text that there are references to work with silkworm silk that has been assumed to apply to spider silk. This may not necessarily be the case, as the proteins (spidroins vs fibroins) are quite different, with vastly different MWs, compositions (e.g. Tyrosine is rare in spider silk but common in silkworm silk), and behaviours (e.g. greater toughness and supercontraction in spider silks). The author should point out where they have made such assumptions, and make judgements about how applicable they may be.
Lastly, there are just a few misspellings (clavate) and grammatical errors within the paper that should be rectified before resubmission.
Author Response
This paper reviews the literature showing the effectiveness of solid-state NMR methods for elucidating the structures of spider silk, a rapidly growing field with some recent exciting discoveries, including many by the author. I didn’t have time to check the outcomes cited against the literature but assume the author, as an authority on the subject, to have done due diligence. I nevertheless do suggest the author check cited datasets and their interpretations for accuracy. I have identified nonetheless some broader issues that I request the author address regarding assumptions about spider silk diversity, silk properties, structures, and variability thereof.
Firstly, the author concentrates the review almost exclusively on major ampullate silk. This is because it is the silk for which the vast majority of ssNMR has been done for. This is probably because of the volumes needed for ssNMR are only attainable with MA silk, and even then for particularly large orb weaving spiders such as Nephila spp. This is fine, but the author should acknowledge this and state that little is known of the other silks. However, as the techniques evolve we may be able to work with other spider silks and/or silks of other species, including smaller orb weavers and non-orb weavers, e.g. RTA clade spiders, cribellate spiders, Mygalamorphs.
Answer:
The author concentrated here to introduce solid state NMR (ss NMR) works about major ampullate of Nephila spp. The reason is as follows. In previous spider silk researches, the remarkable tensile properties of orb web spider dragline silks have attracted a great deal of interest and the dragline silk of the golden-orb weaver, Nephila clavipes has become the benchmark among spider silks. The excellent properties of the spider dragline silk fiber have been interpreted on the basis of the primary and higher order structures of major ampullate of N. clavipes by many researchers because the most abundant fraction of the spider dragline silk is major ampullate.
As the reviewer 2 pointed out, the volumes are necessary for the ss NMR analysis. So, in the future, for instance by using DNP techniques to improve S/N ratio, the characterization of other silks and/or silks of other species, including smaller orb weavers and non-orb weavers, will be possible. Therefore, I emphasized this point in the future aspects of revised review as described below.
On page 2 the authors compared the use of ssNMR with that of SAXS/WAXS and other spectroscopic techniques and concluded NMR to be superior. This is not necessarily true. ssNMR can certainly do things that SAXS/WAXS etc. cannot (e.g. on the fly analyses of amino acid distribution and the relative contributions of alanine and glycine in key secondary structures), but SAXS/WAXS and so on can do other things that NMR cannot (e.g. visualization of the orientation and alignment across the crystalline and amorphous regions). I think a more even-handed comparison is required, showing, where applicable, where NMR has added value to other types of analyses. Jeff Yarger’s work has many examples, which the author would know.
Answer.
The author explained merit of use of ss NMR for the studies about the structure and dynamics of the dragline spider silk more carefully and therefore, Introduction was revised in the revised review as follows.
A variety of techniques such as X-ray diffraction, Fourier transform infrared / Raman spectroscopies and transmission electron microscopy, have been applied to clarify the structure of the spider dragline silk [13-25]. By using these techniques, the structures have been studied from secondary structures to molecular arrangement to hierarchical structure. For example, the X-ray diffraction technique has been used extensively to clarify the structure of silk at the atomic/molecular level, including the size and orientation of nano-crystallites, and also molecular chain arrangement. However, because of the semi-crystalline nature of silk, it is quite difficult to study the atomic/molecular structure of the amorphous region in the silk by X-ray diffraction. To replace or complement this technique, NMR spectroscopy has been applied recently and has been demonstrated as a very effective method to clarify the structure and dynamics of the dragline spider silk [26-65].
As pointed out by the reviewer 2, Jeff Yarger’s group did excellent works about the structure and dynamics of spider silks using several ss NMR techniques. Therefore, I described the NMR techniques briefly in Page 11, lines 253-262 in this review as follows.
Holland and Yarger groups studied the structure and dynamics of stable-isotope labeled spider dragline silks in the dry and hydrated states very actively using several advanced two-dimensional (2D) solid-state NMR i.e., 2D 13C-13C correlation spectrum with dipolar assisted rotational resonance (DARR), 2D 1H-13C Heteronuclear correlation (HETCOR) and 2D 13C-15N HETCOR NMR, 2D incredible natural abundance double quantum transfer experiment (INADEQUATE). Details of their works were introduced in refs [89,90,92].
Actually, Yarger and my groups already reviewed together about recent our studies about silk structure using ss NMR (refs [89,90]). I avoid duplication of contents in this review (Page 2, lines 71-73).
The author has assumed all spider silk to be represented by Nephila clavipes/clavata MA silks. That there are other silks that differ from MA silks I commented on above (so addressing that comment will suffice). However even MA silk is not homogeneous in structure and function. A single species, even individual, spider’s MA silk has a high degree of variability across spinning conditions (free fall vs forces vs anesthetized/unaethetized), diets (Blamires’, C. Craig, Tso’s papers), geography (Blamires/Tso), microhabitat, and exposure to toxins (e.g. insecticides) etc (reviewed by Blamires et al. 2017 Ann Rev Entomol). To date, no good NMR studies have been done to assess this variability at a nano-scale (although some good SAXS/WAXS work has been done), but the potential exists for studies to be done, for instance by using DNP
techniques to smooth out the noise. I encourage the author to acknowledge the types and causes of variability and touch on the possibilities for assessing it using NMR.
Answer.
As the reviewer 2 pointed out, it is interesting to apply ss NMR to other silks and/or silks of other species, including smaller orb weavers and non-orb weavers using DNP techniques to improve S/N ratio. Therefore, I emphasized this point in the future aspects of this review together with citation of the recent review by S.J.Blamires et al., Ann. Rev. Entomol., 2017, 62, 443-460 in this review as follows.
Moreover, it is important to accumulate a lot of knowledge about the structures and dynamics of other silks and/or silks of other species, including smaller orb weavers and non-orb weavers, and also change in the structure and dynamics of the silks to explain spider silk property variations in ecological and evolutionary contexts other than hydration [114]. NMR spectroscopy, especially ss NMR including Dynamic Nuclear Polarization (DNP) [115] will undoubtedly use more frequently for the purpose.
I have noticed by tracking the literature cited within the text that there are references to work with silkworm silk that has been assumed to apply to spider silk. This may not necessarily be the case, as the proteins (spidroins vs fibroins) are quite different, with vastly different MWs, compositions (e.g. Tyrosine is rare in spider silk but common in silkworm silk), and behaviours (e.g. greater toughness and supercontraction in spider silks). The author should point out where they have made such assumptions, and make judgements about how applicable they may be.
Answer
The comparison of the structure and dynamics including the mechanical properties of B. mori silk and spider dragline silk is very interesting and very important. Therefore, a number of studies has been published. Many interesting information has been obtained using ss NMR taking advantage of its characteristics (refs [89,90,92]). The study about Tyr residue in the spider silk is also important as reported by Yarger’s group (ref.49) although the content of Tyr residue is small.
Lastly, there are just a few misspellings (clavate) and grammatical errors within the paper that should be rectified before resubmission.
Answer
Reviewer 1 kindly checked the misspellings and grammatical errors in this review and therefore we revised them.